# Spectrum Extension of a Real-Aperture Microwave Radiometer Using a Spectrum Extension Convolutional Neural Network for Spatial Resolution Enhancement

Guanghui Zhao [ID], Yuhang Huang, Chengwang Xiao *, Zhiwei Chen [ID] and Wenjing Wang

Science and Technology on Multi-Spectral Information Processing Laboratory, School of Electronic Information and Communications, Huazhong University of Science and Technology, Wuhan 430074, China; guanghui_zhao@hust.edu.cn (G.Z.); yuhang_huang@hust.edu.cn (Y.H.); d201780536@hust.edu.cn (Z.C.); m202072171@hust.edu.cn (W.W.)
* Correspondence: d201880571@hust.edu.cn

**Abstract:** Enhancing the spatial resolution of real-aperture microwave radiometers is an essential research topic. The accuracy of the numerical values of brightness temperatures (BTs) observed using microwave radiometers directly affects the precision of the retrieval of marine environmental parameters. Hence, ensuring the accuracy of the enhanced brightness temperature values is of paramount importance when striving to enhance spatial resolution. A spectrum extension (SE) method is proposed in this paper, which restores the suppressed high-frequency components in the scene BT spectrum through frequency domain transformation and calculations, specifically, dividing the observed BT spectrum by the conjugate of the antenna pattern spectrum and applying a Taylor approximation to suppress error amplification, thereby extending the observed BT spectrum. By using a convolutional neural network to correct errors in the calculated spectrum and then reconstructing the BT through inverse fast Fourier transform (IFFT), the enhanced BTs are obtained. Since the extended BT spectrum contains more high-frequency components, namely, the spectrum is closer to that of the original scene BT, the reconstructed BT not only achieves an enhancement in spatial resolution, but also an improvement in the accuracy of BT values. Both the results from simulated data and satellite-measured data processing illustrate that the SE method is able to enhance the spatial resolution of real-aperture microwave radiometers and concurrently improve the accuracy of BT values.

**Keywords:** real-aperture microwave radiometer; spectrum extension; spatial resolution enhancement; land-to-sea contamination; neural network

## 1. Introduction

Compared to infrared, microwaves possess the ability to penetrate through clouds [1] and fogs [2,3]. When cloud cover interference affects infrared measurements, passive microwave measurements can offer more comprehensive sea surface temperature information [4]. However, microwave measurements exhibit lower spatial resolution compared to infrared measurements [5–7]. For instance, the spatial resolution of the infrared radiometer MODIS [8] is superior to 1 km, whereas various observation channels of the microwave radiometer SMR [9] achieve spatial resolutions optimally around 19 km and, at their worst, even as low as approximately 100 km.

A significant issue is brought about by low spatial resolution: land-to-sea radiation contamination of nearshore observations. This contamination renders the observed data from nearshore areas unusable for the retrieval of oceanic environmental parameters [10], such as wind speed, water vapor, and sea surface temperature. While observing nearshore areas, the antenna's main lobe receives radiation from the coastal land, consequently leading to the contamination of observed brightness temperatures (BTs) by land radiation [11].

Due to the substantial contrast in BTs between land and ocean, land-to-sea radiation contamination introduces considerable errors into oceanic observations. Spatial resolution is defined as the size of the footprint projected onto the Earth's surface from the—3dB beam of the antenna's main lobe. The lower the spatial resolution (i.e., the larger the observed footprint), the more land enters the observation area, intensifying the severity of land-to-sea radiation contamination. Enhancing spatial resolution has the potential to mitigate the impact of land-to-sea radiation contamination on nearshore observations.

Furthermore, when the radiometer's antenna footprint transitions from ocean (low BT) to land (high BT), the land area entering the antenna's main lobe gradually increases, eventually filling the main lobe with land radiance. Throughout this process, observed BT gradually rises, creating a transitional zone of the BT gradient at the land–ocean boundary. Higher spatial resolution leads to a narrower main lobe, and consequently, a narrower transitional zone of BT gradient at the land–ocean boundary.

To enhance the spatial resolution of real-aperture microwave radiometers, scholars have proposed numerous methods. Traditional approaches include data fusion [12,13] and antenna pattern synthesis (Backus-Gilbert method) [14,15]. While these methods have yielded promising results, they also exhibit limitations. Data fusion requires a high degree of correlation between information from different sources, and the BG method demands sufficiently dense antenna observation footprints to achieve satisfactory outcomes. In recent years, deep learning algorithms have also been employed to enhance the spatial resolution of microwave radiometers, surpassing the performance of traditional methods [16]. However, in the study [16], the observed BT from a high-frequency observation channel (89 GHz channel) was used as the training target for the low-frequency observation channel (18.7 GHz channel). Due to the frequency disparity, the corresponding microwave radiance characteristics differ as well. This mismatch between the features of the training data used to train the neural network and the actual measured data that the neural network is intended to process undermines the robustness and generalization capabilities of the trained neural network.

Enhancing the spatial resolution of the observed BT is crucial; however, ensuring the utmost accuracy of the final output BT remains equally essential. This accuracy holds significant importance because the numerical values of BTs directly influence the outcomes of oceanic environmental parameter retrieval. However, among the existing methods used to improve the spatial resolution of real-aperture microwave radiometers, the majority predominantly concentrate on enhancing spatial resolution, with relatively less attention directed towards the accuracy of BT values.

In existing relevant studies, the visibility extension (VE) method [17] has not only improved the spatial resolution of real-aperture synthetic microwave radiometers, but has also reduced the reconstruction error of BTs. Likewise, the cosine visibility extension (CVE) method [18], while enhancing the spatial resolution of mirrored-aperture synthetic microwave radiometers, has also led to a decrease in the reconstruction error of BTs. The present study in this paper is inspired by these methods.

The degradation in the spatial resolution of real-aperture microwave radiometers primarily stems from the smoothing of scene BT by the antenna pattern, which results in the blurring of fine details. From a frequency domain perspective, this effect is equivalent to a low-pass filter that suppresses the high-frequency components of the scene BT spectrum.

Whereupon, the spectrum extension (SE) method is proposed based on deep learning to enhance the spatial resolution of real-aperture microwave radiometers while simultaneously improving the accuracy of observed BT values. For simplicity, the spatial spectrum of the original scene BT is referred to as the "scene spectrum," denoted $S_S$; the spatial spectrum of the observed BT is referred to as the "observed spectrum," denoted $S_O$; and the difference between the scene spectrum and the observed spectrum is referred to as the "difference spectrum," denoted $S_D$. The calculated value of $S_S$ through frequency domain transformation and computation of the observed BT and antenna pattern is denoted $S_S^{'}$, representing the extended spectrum. Subtracting $S_O$ from $S_S^{'}$ yields the calculated value of

the difference spectrum, denoted $S_D'$ (i.e., $S_D' = S_S' - S_O$). Using $S_D'$ and $S_D$, a dataset is established and a neural network is trained to learn the mapping relationship $f : S_D' \rightarrow S_D$. The SE method takes the input $S_D'$ and feeds it into the trained neural network. After undergoing processing by the neural network, the estimated value of $S_D$ is obtained, denoted $S_D''$. Adding $S_D''$ to $S_O$ yields the estimated value of the scene spectrum, denoted $S_S''$ (i.e., $S_S'' = S_D'' + S_O$). Applying the inverse fast Fourier transform (IFFT) to $S_S''$ yields the estimated value of the original scene BT, i.e., the processed BT. As a result of the SE method, the BT spectrum becomes closer to the original scene's spectrum. Therefore, the processed BT not only gains an improved spatial resolution, but also achieves BT values closer to the true values of the original scene.

The key points of the SE method are as follows: (1) The expression of observed BT is restructured into a form suitable for convolution integration, facilitating Fourier transformation and computation of the scene spectrum. (2) A Taylor approximation is employed during the computation of the scene spectrum to mitigate error amplification. (3) A neural network is utilized to learn the mapping relationship between the calculated value ($S_D'$) and the ideal value ($S_D$) of the difference spectrum, thus correcting errors present in $S_D'$.

The performance of the SE method is evaluated using both simulation data and satellite observation data. The satellite observation data are obtained from the scanning microwave radiometer (SMR) carried on board the Haiyang-2B (HY-2B) satellite [9]. The SMR operates in a conical scan mode. Its observation frequencies are at 6.925, 10.75, 18.75, 23.85, and 37 GHz. Among these, the 23.85 GHz frequency has only one vertical polarization (V-pol) channel, while the other frequencies have both vertical and horizontal polarization (H-pol) channels [9]. For simplicity, in this paper, the $f$ GHz V-pol or H-pol channels are denoted $f$-V or $f$-H, respectively, such as 6.925-V and 37-H. Because the spatial resolution of channel at 6.925 GHz is lower than that at other frequencies SMR, the demand to enhance spatial resolution at 6.925 GHz is more urgent than that at other frequencies. Therefore, the observation data from the 6.925-H channel (the V-pol channel would be all right as well) of SMR are selected in this paper to assess the performance of the SE method.

## 2. Spectrum Extension (SE) Method

Taking the observations from SMR aboard HY-2B [9] as an example, a schematic is illustrated in Figure 1. In the diagram, $O$ represents a point on the flight track of SMR, while $O'$ represents the nadir point corresponding to $O$. The z-direction is defined along the line $OO'$, and the satellite's flight direction is designated as the y-direction, establishing a right-handed coordinate system $O$-$xyz$. $P$ signifies the observation point, which is the center of the observation footprint. Once point $O$ is selected, the $O$-$xyz$ coordinate system remains fixed and does not move or rotate with the flight of SMR or the scanning of its antenna.

The observed BT from SMR, $T_G$, can be represented by (1) [19].

$$T_G(\alpha, \beta) = \int_0^{2\pi} \int_0^{\pi} T_s(\theta, \varphi) G(\theta, \varphi; \alpha, \beta) \sin\theta d\theta d\varphi \tag{1}$$

where ($\alpha$, $\beta$) represents the observation direction (from $O$ to $P$), ($\theta$, $\varphi$) represents the integration direction (with $\theta$ as the elevation angle and $\varphi$ as the azimuth angle), $T_s(\theta, \varphi)$ denotes the scene BT, and $G(\theta, \varphi; \alpha, \beta)$ stands for the antenna pattern when the observation direction is ($\alpha$, $\beta$). The antenna pattern $G(\theta, \varphi)$ satisfies the normalization condition, as shown in (2).

$$\iint\limits_{4\pi} G(\theta, \varphi) d\Omega = 1 \tag{2}$$

where $d\Omega = \sin\theta d\theta d\varphi$.

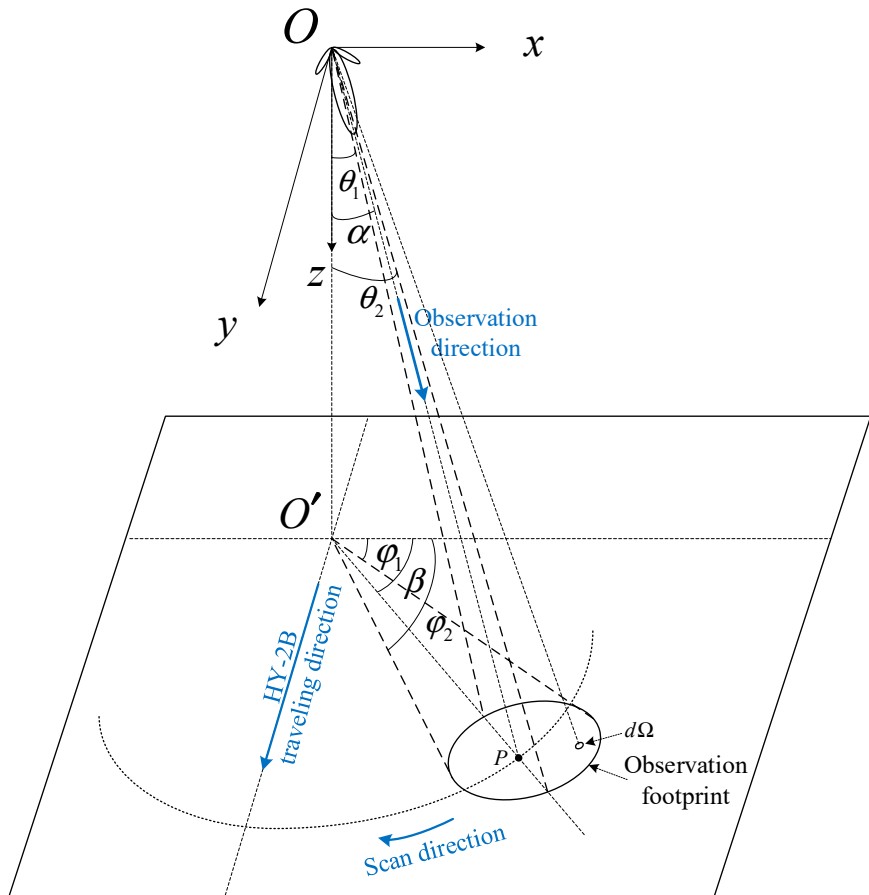

**Figure 1.** Schematic diagram of SMR observation.

From (1) and (2), it can be seen that the output BT from SMR observations results from a weighted average of the antenna pattern over the original scene BT. This weighted average serves to smooth the data, consequently reducing the spatial resolution of the BT; moreover, a wider antenna main lobe leads to a more pronounced decline in spatial resolution. In the frequency domain, the role of the antenna is analogous to that of a low-pass filter. The measurement process resembles a low-pass filtering operation, as it suppresses the high-frequency components of the scene spectrum. Consequently, the spatial resolution of the BT is reduced.

If the high-frequency components suppressed by the low-pass filtering in the scene spectrum can be restored (i.e., the spectrum of observed BT is extended), then reconstructing the scene BT using the restored scene spectrum leads to BT with higher spatial resolution and more precise values. As mentioned above, $S_S$ represents the error-free ideal scene spectrum, while $S_S'$ represents the computed restored scene spectrum, which contains significant errors.

In this paper, by subtracting the observed spectrum ($S_O$) from the scene spectrum, the neural network is trained to learn the mapping relationship of the difference spectrum, resulting in significant improvements. Specifically, $S_S'$ is subtracted by $S_O$ to obtain $S_D'$, which serves as the training input. $S_S$ minus $S_O$ is used as the training target. The neural network is then trained to learn the mapping relationship $f: S_D' \to S_D$, which corrects errors within $S_D'$ and guides the neural network's output $S_D''$ to approximate $S_D$. Upon adding $S_D''$ to $S_O$, the estimated scene spectrum $S_S''$ is obtained, so as to reconstruct the BT image closer to the real scene, that is, the BT image with a higher spatial resolution and more accurate BT values.

In summary, the overall workflow of the SE method is illustrated in Figure 2. The SE method comprises the following five steps:

1. Recovery of Scene Spectrum: By performing fast Fourier transform (FFT) on the antenna pattern $G$ and the observed BT $T_G$, followed by further computations, the calculated value of the scene spectrum $S'_S$ is obtained;
2. Calculation of Difference Spectrum: $S'_S$ is subtracted from the observed spectrum $S_O$ to obtain the calculated value of the difference spectrum $S'_D$;
3. Error Calibration with Neural Network: $S'_D$ is fed into a neural network, which outputs the estimated value of the difference spectrum, $S''_D$;
4. Acquiring Estimation of Scene Spectrum: $S''_D$ is added to the observed spectrum $S_O$ to yield the estimated value of the scene spectrum, $S''_S$;
5. BT Reconstruction: Applying IFFT to $S''_S$ results in the reconstructed scene BT, $T''_S$.

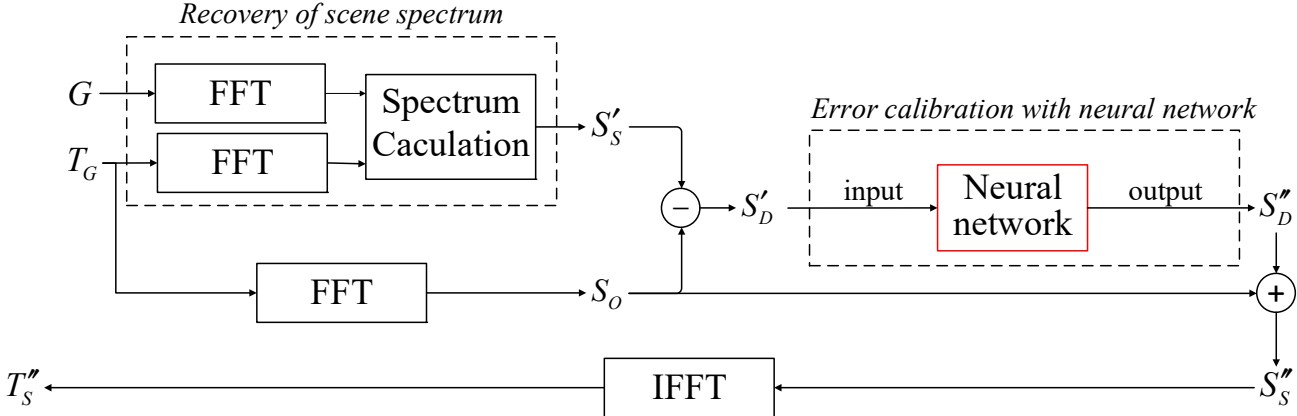

**Figure 2.** Workflow of the SE Method.

In the SE method, the third step is the most crucial. To complete this step, it is necessary to pre-generate a dataset, design a neural network, and carry out network training. For a single-channel real-aperture microwave radiometer system, this process needs to be carried out only once for dataset generation, network design, and training. However, for a multi-channel real-aperture microwave radiometer, this process of dataset generation, network design, and training needs to be repeated for each channel. Subsequently, the trained network can be directly employed to process observational data and enhance spatial resolution.

Next, let us delve into the key techniques within the SE method.

### 2.1. Recovery of Scene Spectrum

In the SE method, the process involves performing a Fourier transform on the observed BT to obtain the observed spectrum. By utilizing the relationship between the scene spectrum (unknown), observed spectrum (known), and the antenna pattern spectrum (known), the scene spectrum is calculated, thus recovering the high-frequency components, to extend the observed spectrum.

Because the original expression of the observed BT is not strictly in the form of convolution integration, performing frequency domain transformation can be quite complicated. Therefore, the SE method employs appropriate approximations to render the expression of the observed BT in line with the convolution integration format. This adjustment facilitates the process of frequency domain transformation.

In order to mitigate the amplification of errors during the calculation of the restored scene spectrum, the SE method employs the Taylor approximation technique. If the scene spectrum were calculated directly, it would involve dividing the observed spectrum (numerator) by the conjugate of the antenna pattern spectrum (denominator). Given that the observed BT and antenna pattern contain measurement errors and noise, errors can be significantly amplified when the denominator's value is very small [20]. By using the Taylor approximation, it becomes possible to restore the scene spectrum by multiplying

the observed spectrum by an approximated term. This approach effectively suppresses the amplification of errors.

### 2.1.1. Fourier Transform of Observed Brightness Temperature

During antenna scanning, only the observation direction changes, while the antenna pattern remains constant. Hence, (1) can be rewritten as follows:

$$T_B(\alpha, \beta) = \int_0^{2\pi} \int_0^{\pi} T_s(\theta, \varphi) G(\theta - \alpha, \varphi - \beta) sin\theta d\theta d\varphi \tag{3}$$

Neglecting the influence of antenna sidelobes, we have

$$T_G(\alpha, \beta) \approx \int_{\varphi_1}^{\varphi_2} \int_{\theta_1}^{\theta_2} T_s(\theta, \varphi) G(\theta - \alpha, \varphi - \beta) sin\theta d\theta d\varphi \tag{4}$$

where $T_G(\alpha, \beta)$ represents the BT observed in the antenna main lobe, and the integration domain $\Omega_m$ corresponds to the main lobe region, expressed as $\Omega_m = \{(\theta, \varphi)|\theta_1 < \theta < \theta_2, \varphi_1 < \varphi < \varphi_2\}$.

When the main lobe is sufficiently narrow, the variation range of $\theta$ within $\Omega_m$ is small. In this case, the above equation can be approximated as follows:

$$T_G(\alpha, \beta) \approx sin\theta_0 \cdot \int_{\varphi_1}^{\varphi_2} \int_{\theta_1}^{\theta_2} T_s(\theta, \varphi) G(\theta - \alpha, \varphi - \beta) d\theta d\varphi \tag{5}$$

where $\theta_0$ is a constant, expressed as $\theta_0 = (\theta_1 + \theta_2)/2$.

The Fourier transform of (5) is given by

$$S_O(u, v) \approx S_S(u, v) \cdot \overline{S_G(u, v)} \cdot sin\theta_0 \tag{6}$$

where $S_O(u, v) = \mathcal{F}\{T_G(\alpha, \beta)\}$, $S_S(u, v) = \mathcal{F}\{T_s(\theta, \varphi)\}$, and $S_G(u, v) = \mathcal{F}\{G(\theta, \varphi)\}$; here, $\mathcal{F}\{\ \}$ represents the Fourier transform, and the overline denotes the complex conjugate.

### 2.1.2. Calculation of Scene Spectrum

According to (6), the scene spectrum can be restored using (7).

$$S_S'(u, v) = S_O(u,v) \big/ \overline{S_G(u,v)}\, sin\theta_0 \tag{7}$$

Taking into account the errors during scene spectrum restoration, the restored scene spectrum can be represented as follows:

$$S_S'(u, v) = S_S(u, v) + \Delta e(u, v) \tag{8}$$

where $S_S(u, v)$ is the scene spectrum (error-free), $S_S'(u, v)$ is the restored scene spectrum (including errors), and $\Delta e(u, v)$ represents the errors in the restored scene spectrum.

Hence, utilizing the inverse Fourier transform, the reconstruction of scene BT can be expressed as

$$T_s'(\theta, \varphi) = \mathcal{F}^{-1}\{S_S(u, v)\} + \mathcal{F}^{-1}\{\Delta e(u, v)\} \tag{9}$$

where $T_s'(\theta, \varphi)$ is the reconstructed scene BT with a higher spatial resolution (compared to the observed BT $T_G$), $\mathcal{F}^{-1}\{\ \}$ represents the inverse Fourier transform, and $\mathcal{F}^{-1}\{\Delta e(u, v)\}$ denotes the error in reconstructed BT. In an ideal scenario, when $\Delta e(u, v) = 0$, leading to $\mathcal{F}^{-1}\{\Delta e(u, v)\} = 0$, $T_s'(\theta, \varphi)$ would exactly correspond to the scene BT $T_s(\theta, \varphi)$.

However, in reality, certain frequency components in the denominator $\overline{S_G(u, v)}$ of (7) can be extremely small. Dividing by such a small value would lead to the amplification of errors and noise within the numerator $S_O(u, v)$, resulting in a significant increase in $\Delta e(u, v)$ [20]. This, in turn, would cause intolerable errors in $T_s'(\theta, \varphi)$. Therefore, in the

SE method, during the computation of $T'_s(\theta, \varphi)$, a Taylor series expansion is employed to approximate the division as multiplication. This approximation helps suppress the amplification of errors.

2.1.3. Taylor Approximation for Error Amplification Suppression

Expanding the expression $1/\overline{S_G(u,v)}$ in (7) using a Taylor series (or a MacLaurin series) approximation yields (10):

$$\frac{1}{\overline{S_G(u,v)}} \approx 1 + (1 - \overline{S_G(u,v)}) + (1 - \overline{S_G(u,v)})^2 \\ + (1 - \overline{S_G(u,v)} + \cdots + (1 - \overline{S_G(u,v)})^r \tag{10}$$

where *r* represents the order of the Taylor series expansion, with a convergence condition of $\left|1 - \overline{S_G(u,v)}\right| < 1$. Given that $G(\theta, \varphi)$ is normalized through (2), $0 < \overline{S_G(u,v)} \le 1$ is satisfied within the convergence region. After applying the approximation in (10), (7) can be rewritten as (11), so as to effectively suppress the error amplification in $T'_s(\theta, \varphi)$.

$$S'_S(u,v) \approx \frac{S_O(u,v)}{\underline{sin\theta_0}}[1 + (1 - \overline{S_G(u,v)}) + (1 - \overline{S_G(u,v)})^2 \\ + (1 - S_G(u,v)^3 + \cdots + (1 - \overline{S_G(u,v)})^r] \tag{11}$$

However, if the direct inverse Fourier transform is applied to $S'_S(u,v)$ obtained from (11), significant errors still persist in the reconstructed scene BT $T'_s(\theta, \varphi)$ (as seen in (9)). Therefore, the SE method employs a convolutional neural network to correct the errors within $S'_S(u,v)$.

*2.2. Dataset Generation Steps*

The generation of the dataset has a significant impact on the final performance of the neural network. When creating the dataset, using $S_S$ directly as the training target can lead to increased convergence difficulty for the network and result in suboptimal performance. To mitigate this challenge, the computed values ($S'_D$) and the ideal values ($S_D$) of the difference spectrum between $S_S$ and $S_O$ are employed as training inputs and targets for the neural network, which has yielded favorable results. The process for generating the dataset is illustrated in Figure 3, and consists of the following five steps.

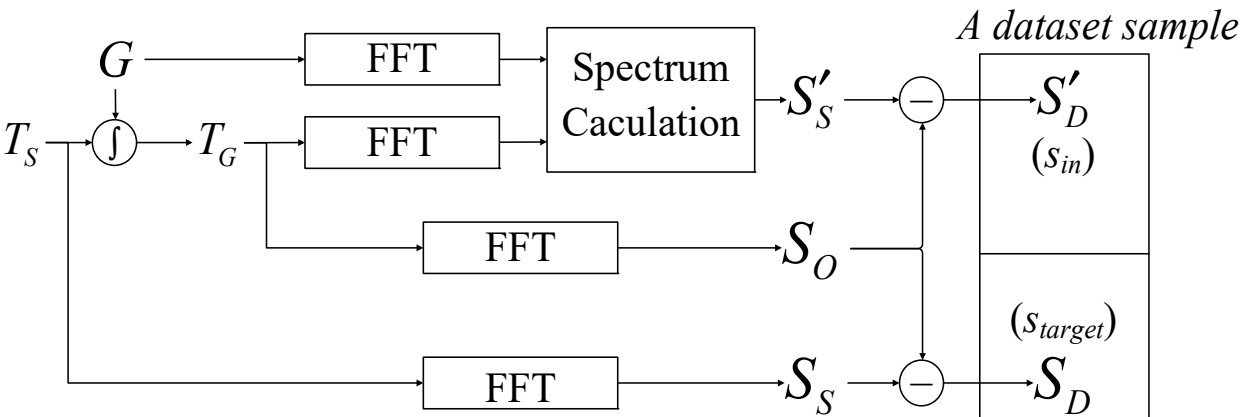

**Figure 3.** Dataset generation process.

1.  An appropriate original scene, $T_s(\theta, \varphi)$, is chosen according to specific circumstances. Specifically, $T_s(\theta, \varphi)$ is selected from observed BT data obtained from a spaceborne microwave radiometer with sufficiently high spatial resolution. Then, the observed BT, $T_G(\alpha, \beta)$, is generated using (4).

2. FFT is performed on $T_s(\theta, \varphi)$ and $T_G(\alpha, \beta)$ to obtain the scene spectrum, $S_S(u, v)$, and the observed spectrum, $S_O(u, v)$.

3. According to (11), an appropriate value for $r$ (the order of the Taylor series expansion) is selected, and $S'_S(u, v)$ is computed.

4. $S'_D(u, v)$ is computed by subtracting $S_O(u, v)$ from $S'_S(u, v)$, and an $n \times n$ matrix is extracted from $S'_D(u, v)$ as the training input, $s_{in}$. Similarly, $S_D(u, v)$ is computed by subtracting $S_O(u, v)$ from $S_S(u, v)$, and an $n \times n$ matrix is extracted from $S_D(u, v)$ as the training target, $s_{target}$. Together, $s_{in}$ and $s_{target}$ form a dataset sample.

5. The above four steps are repeated to generate multiple dataset samples. These samples collectively constitute the dataset.

In the third step, the selection of the value of $r$ directly affects the convergence outcome of the neural network, and consequently influences the final results. Additionally, when generating the dataset, if the Taylor expansion approximation is not used, the process in the third step involves directly calculating $S'_S(u,v)$ based on (7), while keeping the remaining steps unchanged. The impact of different sets of $r$ values on the convergence of neural network training is investigated, and it is found that an $r$ value of 60 yields better results than other values. Therefore, unless otherwise stated, r is valued at 60 in the third step of dataset generation.

In the fourth step, during the generation of $s_{target}$ and $s_{in}$, the subtraction of the observed spectrum $S_O(u,v)$ from the scene spectrum $S_S(u,v)$ and its computed value $S'_S(u,v)$ is performed. This approach enhances the convergence of the neural network during training, resulting in faster and better convergence.

*2.3. SE-CNN*

The difference spectrum signal is, in fact, a spatial frequency spectrum signal, and its distribution characteristics bear some resemblance to the visibility signal [17] and the cosine visibility signal [18]. Consequently, similar to the VE method and the CVE method, the neural network model employed in the SE method is predominantly composed of CNN architectures [21,22].

The main task of the neural network in the SE method is to learn the mapping relationship between the calculated values and the ideal values of the difference spectrum, i.e., $f : S'_D \to S_D$. The calculated difference spectrum, which includes a significant amount of computational error, is served as the input signal. Therefore, when designing the network architecture, the initial step involves a fully connected mapping of the input signal $S'_D$, which provides an initial correction for the errors within the input signal. Following this, convolutional operations are performed to further correct these errors. For the sake of convenience, the neural network model employed in the SE method described in this paper is referred to as SE-CNN.

2.3.1. Network Architecture

The neural network model (i.e., SE-CNN) designed in this paper consists of a main branch and a side branch, as illustrated in Figure 4. The main branch comprises two fully connected layers (FC1, FC2), a dropout layer, and three convolutional layers (C1, C2, C3). The side branch includes three convolutional layers (BC1, BC2, BC3). The outputs of the main branch and side branch are summed to obtain the final output of the neural network.

The input to the neural network is a complex matrix of size $n \times n$. With its real and imaginary parts each represented by an $n \times n$ matrix, it is represented as a $2 \times n \times n$ matrix, denoted $\boldsymbol{P}$.

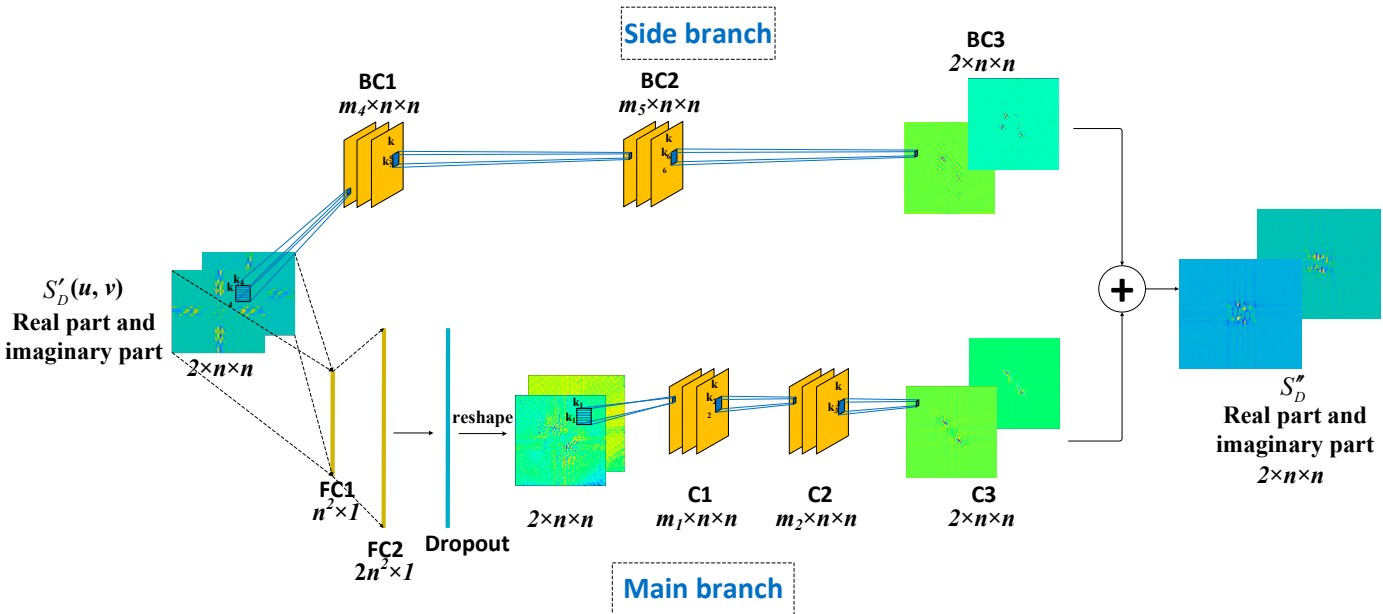

**Figure 4.** Network architecture of SE-CNN.

In the main branch, all elements of matrix $P$ are connected to a fully connected layer FC1 with $n^2$ nodes, followed by activation using the parametric rectified linear unit (PReLU) function. The expression for the PReLU function, denoted (12), is as follows:

$$\mathrm{PReLU}(x) = \begin{cases} x, & x \geq 0 \\ ax, & \text{otherwise} \end{cases} \tag{12}$$

where $a$ is a learnable parameter, and its value is continuously updated during the training process. The output of FC1 layer is fully connected to another fully connected layer FC2 with $2n^2$ nodes. The output of FC2 is then activated using the PReLU function. A dropout layer [23] is applied to the output of FC2 to prevent overfitting. The output matrix of the dropout layer (with dimensions $2n^2 \times 1$) is reshaped to form a $2 \times n \times n$ matrix, which is then connected to convolutional layers C1-C3 for further convolution operations. C1 contains $m_1$ convolutional kernels of size $k_1 \times k_1$, and C2 contains $m_2$ convolutional kernels of size $k_2 \times k_2$. Both convolution operations have a stride of 1 and are followed by batch normalization (BN) [24] and activation using the PReLU function. C3 consists of two convolutional kernels of size $k_3 \times k_3$, and its output is a $2 \times n \times n$ matrix, which serves as the final output of the main branch.

In the side branch, $P$ is directly fed into convolutional layers BC1-BC3. BC1 consists of $m_4$ filters of size $k_4 \times k_4$, and BC2 consists of $m_5$ filters of size $k_5 \times k_5$, both followed by a tanh activation function, with a convolutional stride of 1. BC3 contains two filters of size $k_6 \times k_6$, and its output is a $2 \times n \times n$ matrix, which is the output of the side branch.

After summing the outputs of the main branch and the side branch, a $2 \times n \times n$ matrix is obtained. This matrix is divided into two $n \times n$ matrices, corresponding to the real and imaginary parts of $S_S''$. These matrices constitute the final output of the SE-CNN.

### 2.3.2. Training Process

The training process of SE-CNN is illustrated in Figure 5.

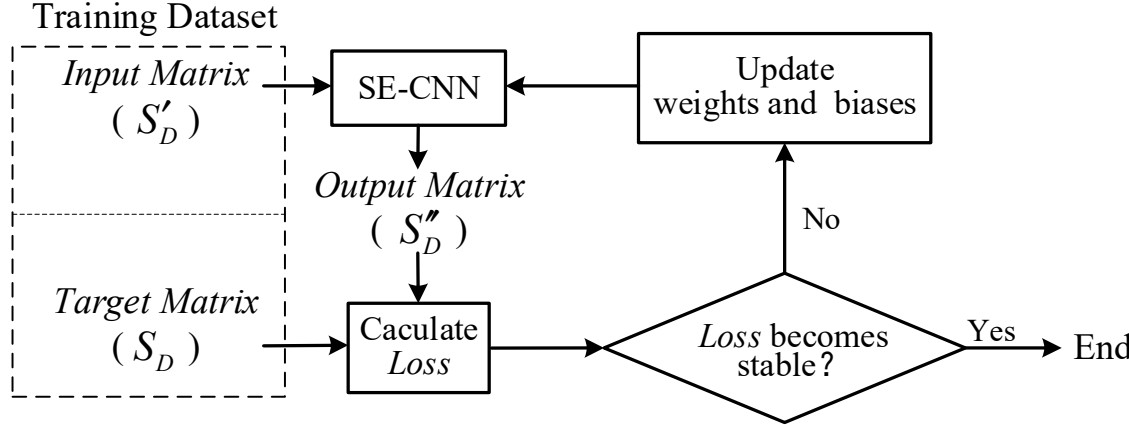

**Figure 5.** Training process.

During the training of SE-CNN, a batch of samples is selected from the training dataset, and the input matrices $S'_D$ from this batch of samples are fed into the SE-CNN. Subsequently, the output matrices $S''_D$ are compared with the target matrices $S_D$ of this batch of samples. The mean squared error is then calculated using (13), which serves as the loss function (*Loss*).

$$Loss = \frac{1}{N_b} \frac{1}{2n^2} \sum_{s=1}^{N_b} \sum_{m=1}^{n^2} \left[ \left| \mathrm{real}\left( S''_{D,\,s}(m) - S_{D,s}(m) \right) \right|^2 + \left| \mathrm{imag}\left( S''_{D,\,s}(m) - S_{D,s}(m) \right) \right|^2 \right] \tag{13}$$

where $N_b$ is the number of batches of samples, $s$ represents the sample index, $n^2$ is the total number of elements in the complex matrix $S_{D,s}$ (or $S''_{D,\,s}$), $m$ denotes the element index within the matrix, $S''_{D,\,s}$ is the output complex matrix corresponding to the s-th sample, $S_{D,s}$ is the target complex matrix for the s-th sample, real$(\cdot)$ represents the real part of a complex number, imag$(\cdot)$ represents the imaginary part of a complex number, and $|\cdot|$ denotes the absolute value.

After computing the loss function, the weights and biases of the neural network model are updated using the Adam algorithm [25]. The Adam algorithm offers several advantages over other stochastic optimization methods. Firstly, it is computationally efficient and requires relatively low memory, making it suitable for problems involving a large amount of data and/or parameters. Secondly, it is invariant to diagonal rescaling of gradients, which helps enhance performance on ill-conditioned problems. Thirdly, it is applicable to non-stationary objectives and problems with highly noisy and/or sparse gradients, which are common in machine learning applications. Fourthly, the hyperparameters of the Adam algorithm have intuitive interpretations and usually require less tuning.

Subsequently, another batch of samples from the training dataset is input to SE-CNN for further training. When all the samples in the training dataset have been used for one round of training, an epoch is completed. Then, the training proceeds to the next epoch.

During the training process, a learning rate scheduler is employed, which adjusts the learning rate based on the number of epochs during training. Specifically, when the loss function does not decrease for 10 consecutive epochs, the learning rate is reduced to a fifth of its original value. Learning rate schedulers are often beneficial for the training of models as they help in optimizing the learning process.

Over multiple epochs of iterative training, the weights and biases of the network model are continuously updated to minimize the loss function. During the training process, as the number of epochs increases, both the training dataset's loss function (referred to as train-loss) and the testing dataset's loss function (referred to as test-loss) typically decrease initially and then stabilize. Training is stopped when the loss functions converge to a stable state. This way, the entire training process finishes.

*2.4. Metrics*

As mentioned above, the spatial resolution is enhanced with the SE method by recovering the scene spectrum (especially high-frequency components). So, the root mean square error (RMSE) of the spectrum relative to scene spectrum, denoted as $R$, is used to assess the degree of scene spectrum recovery, and the corresponding calculation formula is shown in Equation (14).

$$R = \sqrt{\frac{1}{n^2}\sum_{i=1}^{n}\sum_{j=1}^{n}\left(|S_{s,ij}| - |S_{B,ij}|\right)^2} \tag{14}$$

where $R$ represents the RMSE of spectrum to be assessed, $n$ is the number of rows (or columns) in the spectrum data matrix, $S_s$ stands for scene spectrum, $S_B$ denotes the spectrum to be assessed, $i$, $j$ are row and column indices of the spectrum data matrix, and $\|$ means taking the magnitude (absolute value) of a complex number.

Meanwhile, the accuracy of BTs is assessed using the RMSE relative to the original scene's BTs. The RMSE of BTs, denoted $E$, is calculated using (15), as shown below.

$$E = \sqrt{\frac{1}{n^2}\sum_{i=1}^{n}\sum_{j=1}^{n}\left(T_{s,ij} - T_{B,ij}\right)^2} \tag{15}$$

where $E$ represents the RMSE of BTs to be assessed, $n$ is the number of rows (or columns) in the BT data matrix, $T_s$ stands for scene BT, $T_B$ denotes the BT to be assessed, and $i$ and $j$ are the row and column indices of the BT data matrix.

**3. Simulation Data Processing**

The simulation process is essentially the same as the SE method process, as shown in Figure 2. The difference between simulation data processing and the processing of actual satellite data lies in the fact that the observed brightness temperature $T_G$ in simulation is generated through simulation calculations using the simulated scene $T_S$ and the antenna pattern $G$, rather than being generated through observations using onboard radiometers.

*3.1. Dataset Generation*

In order to demonstrate the enhancement in spatial resolution more clearly, it is essential that the spatial resolution of the simulated scene is sufficiently high, and the brightness temperature distribution in the simulated scene should closely approximate that of the actual scene.

When generating the dataset following the steps described in Section 2.2, in Step 1, the observed BTs from the 89.0-GHz (A)-H channel of AMSR2 [26] (L1B product) are chosen as the simulated scene $T_S$. Simultaneously, the antenna main beam pattern of SMR's 6.925-H channel is selected as $G$, as illustrated in Figure 6a. In Step 4, $n = 75$, meaning that both the input and output of the SE-CNN are matrices of size $75 \times 75$.

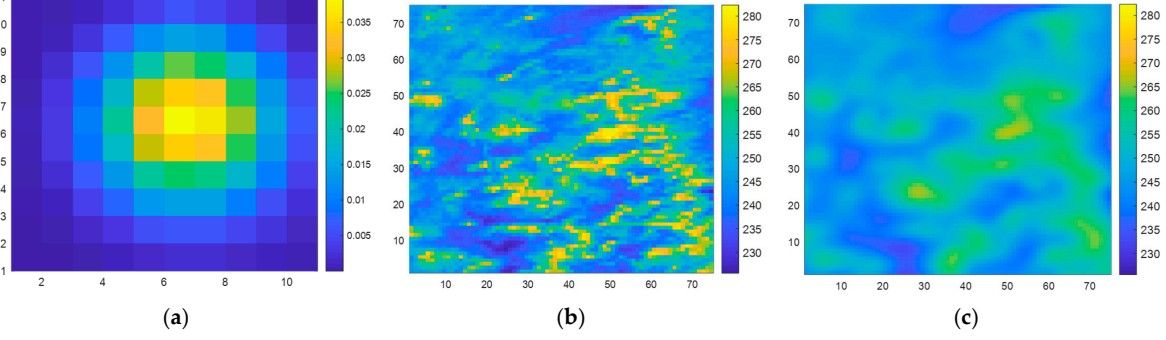

**Figure 6.** An example of scene observation with the scene chosen from 89-GHz AMSR2 BTs: (**a**) antenna main lobe pattern; (**b**) scene chosen from 89-GHz AMSR2 BTs; (**c**) observed BT.

A training dataset of 8000 samples and a testing dataset containing 2000 samples are constructed. Illustrated in Figure 6b,c is an example of the original scene, BT $T_S$, and the observed BT $T_G$. Figure 7 depicts the corresponding sample within the dataset, including the network input ($S'_D$) and target output ($S_D$).

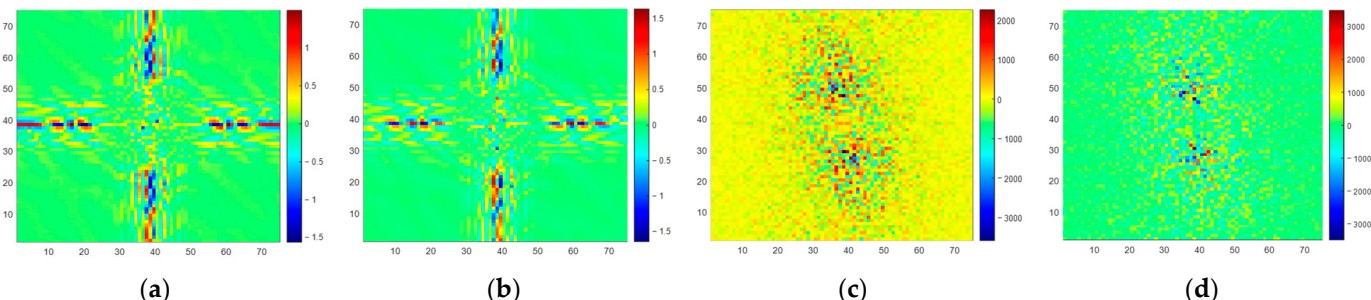

| (a) | (b) | (c) | (d) |

**Figure 7.** An example of network input ($S'_D$) and target output ($S_D$) with the scene chosen from 89-GHz AMSR2 BTs: (**a**) Real part of $S'_D$; (**b**) imaginary part of $S'_D$; (**c**) real part of $S_D$; (**d**) imaginary part of $S_D$.

### 3.2. Hyperparameter Determination

In this subsection, we apply a Bayesian search algorithm [27] to explore more than 40 sets of hyperparameter combinations. To accelerate the hyperparameter search, the Hyperband algorithm [28] serves as an early stopping strategy, halting the training of neural network models that perform poorly. After the search, the hyperparameters that result in the best convergence of the neural network are chosen. Table 1 outlines the network structure parameters. Furthermore, we set the input matrix dimension to $n = 75$, establish a batch size of 64, and employ a $5 \times 5$ kernel for each convolutional layer. The initial learning rate is fixed at 0.094, and we configure the Adam optimizer with parameters $\beta_1 = 0.594$, $\beta_2 = 0.647$, and $\varepsilon = 4.186 \times 10^{-8}$. The corresponding convergence curve of the neural network is displayed in Figure 8.

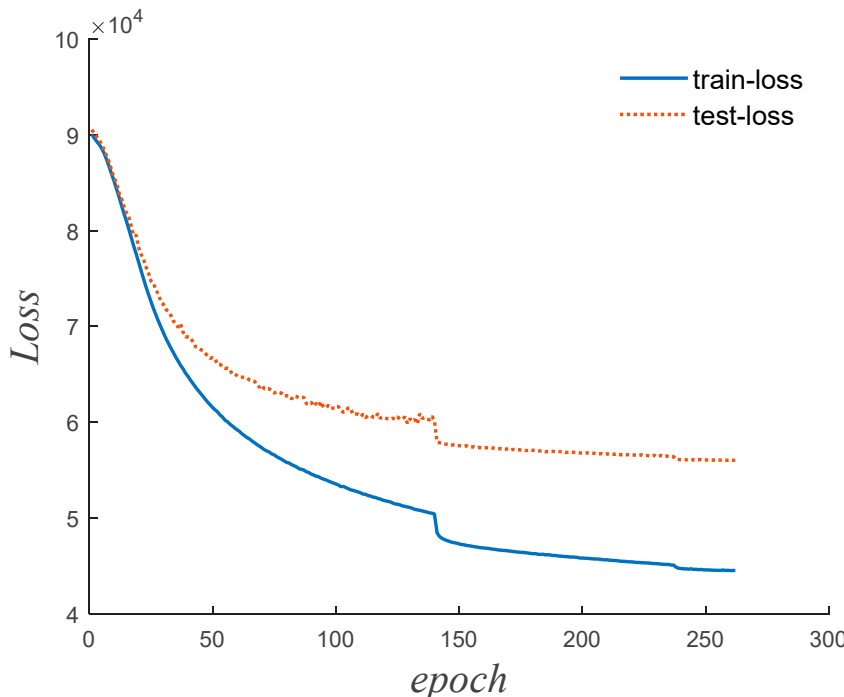

**Figure 8.** Convergence curve of SE-CNN trained by the dataset generated with the scenes chosen from 89-GHz AMSR2 BTs.

**Table 1.** Network structure parameters for SE-CNN trained by the dataset generated with the scenes chosen from 89-GHz AMSR2 BTs.

|  | Layer | Structure Parameter | Value |
|---|---|---|---|
| **Main branch** | FC1 | Node number ($n^2$) | 5625 |
|  | FC2 | Node number ($2n^2$) | 11,250 |
|  | Dropout | Probability (dropout) | 0.159 |
|  | C1 | Kernel number ($m_1$) | 64 |
|  | C2 | Kernel number ($m_2$) | 128 |
|  | C3 | Kernel number | 2 |
| **Side branch** | BC1 | Kernel number ($m_4$) | 64 |
|  | BC2 | Kernel number ($m_5$) | 128 |
|  | BC3 | Kernel number | 2 |

### 3.3. Simulation Data Processing Results

Upon completion of the SE-CNN training, the neural network's performance is assessed using the test dataset samples. Randomly selected test samples are used, and their corresponding test results are presented. As shown in Figure 9, we display three randomly chosen test samples along with their respective test outcomes. In Figure 9a,d,g, the BTs of three simulated scenes (Scene I, Scene II, and Scene III) are depicted, while Figure 9b,e,h showcases the corresponding observed BTs. Furthermore, Figure 9c,f,i illustrate the reconstructed BTs achieved through the SE method, referred to as SE BTs. From Figure 9, it becomes evident that the observed BTs, obtained through a weighted average with the antenna pattern, exhibit blurriness and smoothness, leading to the loss of fine details. This indicates a reduction in spatial resolution. However, after undergoing SE processing, the reconstructed BTs contain more detailed information and closely resemble the original simulated scene's BTs. Consequently, there is an improvement in spatial resolution.

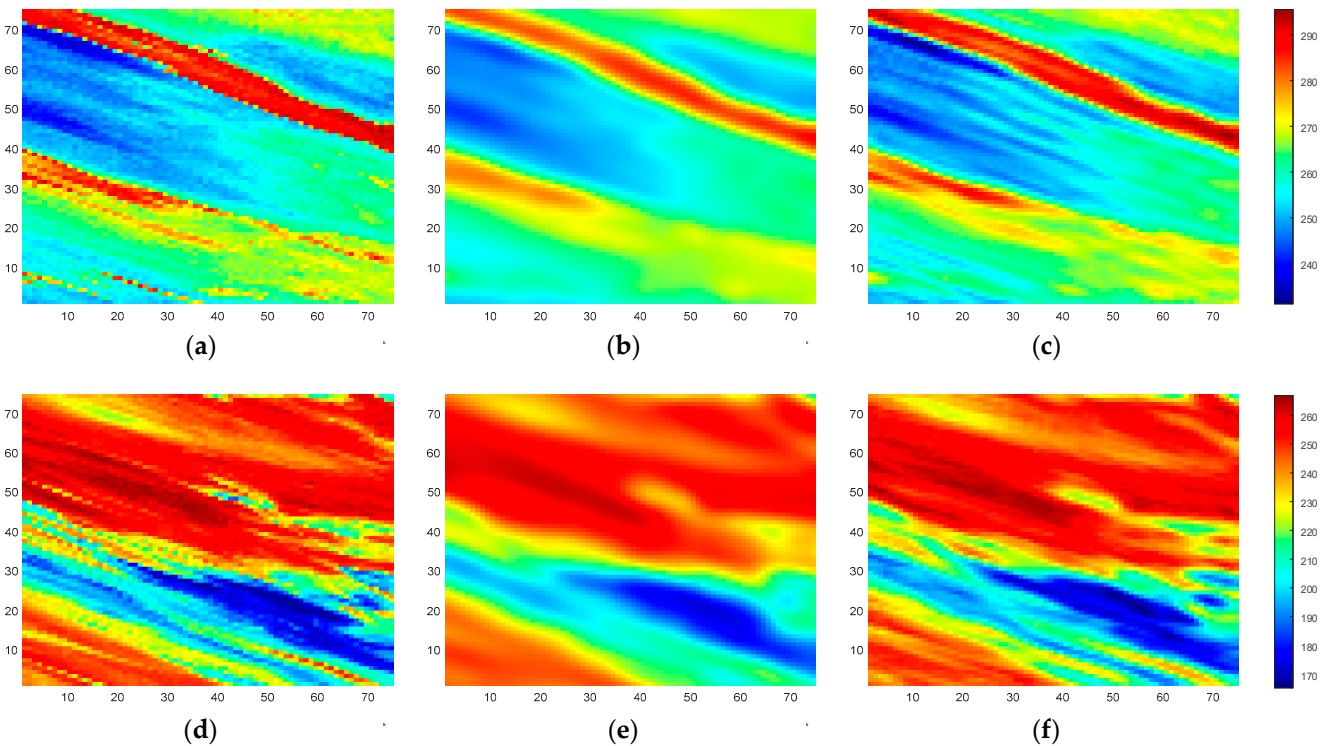

**Figure 9.** *Cont.*

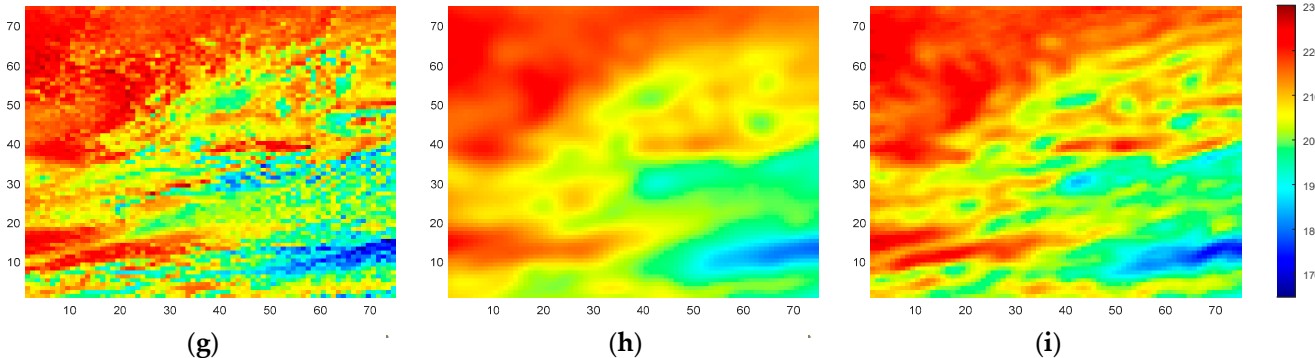

**Figure 9.** Processing results of the test dataset samples. (**a**,**d**,**g**) The BT images of the simulated scenes, Scene I, Scene II, and Scene III; (**b**,**e**,**h**) the corresponding observed BT images; (**c**,**f**,**i**) the corresponding SE BT images.

For a more visually intuitive presentation of the results, we opt to select random cross-sections from the three scenes. Subsequently, the corresponding one-dimensional BT curves are featured in the figures. In Figure 10a, you will find the one-dimensional BT curve extracted from the 45th column of Scene I. Meanwhile, Figure 10b depicts the corresponding curve found at the 43rd column of Scene II, and Figure 10c showcases the curve identified at the 63rd column of Scene III. From the observations made in Figure 10, it becomes evident that the SE BT offers a more precise representation of the original scene's BT distribution when compared to the observed BT. This is particularly noticeable in regions characterized by rapid changes. In summary, the SE BT curve closely aligns with the BT curve of the original scene, indicating that the SE method not only enhances spatial resolution, but also enhances the accuracy of BT values.

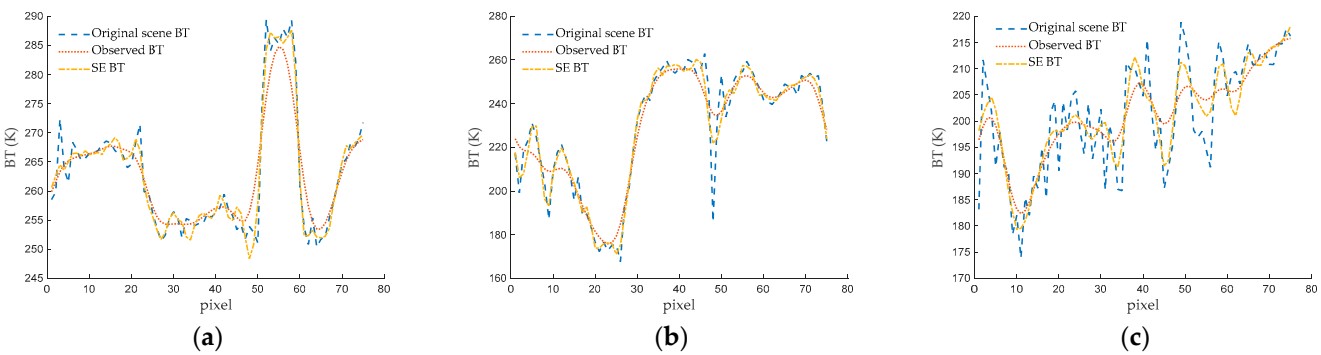

**Figure 10.** One-dimensional BT curves of the three scenes shown in Figure 9. (**a**) Scene I; (**b**) Scene II; (**c**) Scene III.

From a spectrum error perspective, we present the amplitude spectrum error of the observed BT and SE BT in Figure 11 (no discernible pattern was identified in the phase spectrum error; thus, it is not discussed). In Figure 11a,c,e, it is evident that certain spatial frequency components within the observed spectrum ($S_O$) exhibit significant errors. However, in the spectrum of SE BT ($S_S''$) displayed in Figure 11b,d,f, errors in the corresponding spatial frequency components are notably reduced. This observation underscores the SE method's capability to recover spatial frequency components within the scene's spectrum, including high-frequency components. This phenomenon underlines the underlying principle behind the enhancement of spatial resolution achieved through the SE method.

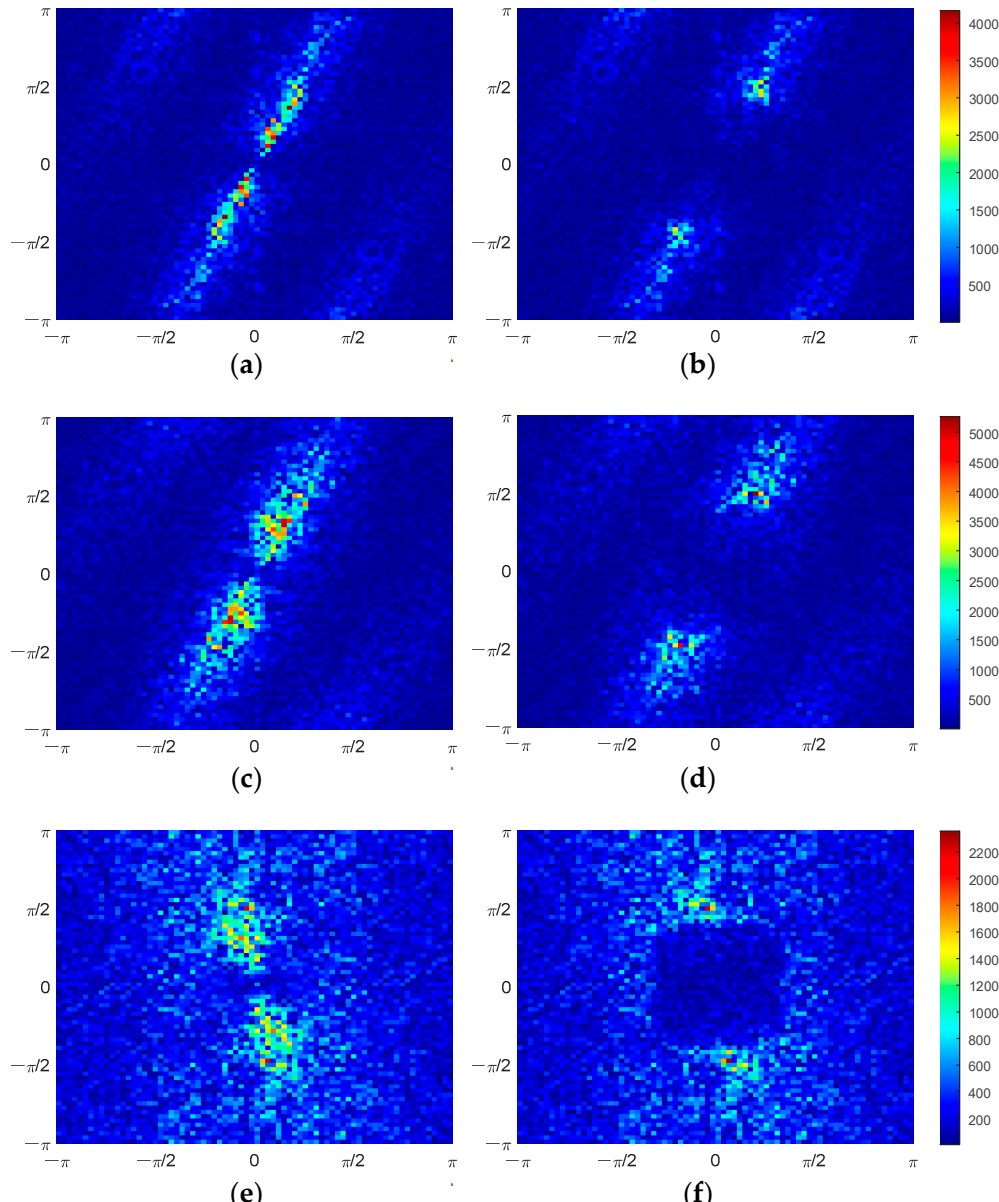

**Figure 11.** Amplitude spectrum error distribution. (**a**,**c**,**e**) Error in $S_O$ of Scene I, Scene II, and Scene III; (**b**,**d**,**f**) Error in $S_S''$ of Scene I, Scene II, and Scene III.

However, it also can be observed from Figure 11 that residual errors still exist in $S_S''$ at higher frequencies. This indicates that the SE-CNN has limited capability in correcting errors in higher-frequency components of $S_S'$. This is also the reason why the spatial resolution of SE BT is lower than that of scene BT. Upon analysis, it is typically the case that as the frequency of scene components increases, their amplitude decreases, sometimes even spanning several orders of magnitude compared to lower-frequency components. When the SE-CNN focuses on correcting errors in larger-amplitude spectrum components, it might inadvertently overlook error correction for smaller-amplitude spectrum components. This is an area that needs to be addressed in the next phase of research.

Concerning the quantitative analysis, we calculate both the spectrum error and BT error. The RMSE for the amplitude spectrum error of $S_O$ is denoted as $R_O$, while the RMSE for the amplitude spectrum error of $S_S''$ is denoted as $R_S''$ (refer to (14)). Following the calculations, $R_O$ for Scene I, Scene II, and Scene III are 366.38, 598.80, and 385.19, respectively. The corresponding $R_S''$ values are 254.28, 428.25, and 331.77. This means reductions of 30.49%, 28.48%, and 14.91%, respectively, as indicated in Table 2. The RMSE

for observed BTs is denoted as $E_O$, while the RMSE for SE BTs is denoted as $E_S''$ (refer to (15)). After the calculations, $E_O$ values for Scene I, Scene II, and Scene III are 4.89 K, 7.98 K, and 5.14 K, respectively. The corresponding $E_S''$ values are 3.34 K, 5.60 K, and 4.40 K. These values represent reductions of 34.70%, 29.82%, and 14.40%, respectively, as presented in Table 3.

**Table 2.** Amplitude spectrum error of observed BT and SE BT for Scene I, Scene II, and Scene III.

|  | $R_O$ | $R_S''$ | Reduction |
|---|---|---|---|
| Scene I | 366.38 | 254.28 | 30.49% |
| Scene II | 598.80 | 428.25 | 28.48% |
| Scene III | 385.19 | 331.77 | 14.91% |

**Table 3.** Error of observed BT and SE BT for Scene I, Scene II, and Scene III.

|  | $E_O$(K) | $E_S''$(K) | Reduction |
|---|---|---|---|
| Scene I | 4.89 | 3.34 | 34.70% |
| Scene II | 7.98 | 5.60 | 29.82% |
| Scene III | 5.14 | 4.40 | 14.40% |

To further validate the effectiveness of the SE method, simulation tests are conducted on all 2000 samples from the test dataset. The average amplitude spectrum error of $S_O$ and $S_S''$ for all test samples is computed, as summarized in Table 4. The average error (refer to (16)) of $S_O$ is 275.84, while the average error of $S_S''$ is 231.52, reflecting a decrease of approximately 16.07%. The average RMSE (refer to (17)) of $S_O$ is 392.66, whereas the average RMSE of $S_S''$ is 307.80, showing a reduction of approximately 21.61%. Additionally, the average error and the average RMSE of observed BT and SE BT for all test samples are calculated and presented in Table 5. The average error (refer to (18)) for both observed and SE BT is approximately 0 K. The average RMSE (refer to (19)) for observed BT is 5.24 K, while the average RMSE for SE BT is 4.05 K, representing a reduction of approximately 22.71%. The improved spatial resolution of SE BT brings it closer to the original BT, resulting in a smaller reconstruction error.

$$\text{A} = \frac{1}{2000} \sum_{p=1}^{2000} \frac{1}{n^2} \sum_{i=1}^{n} \sum_{j=1}^{n} \left( S_{S,p,ij} - S_{B,p,ij} \right) \tag{16}$$

where $p$ is the sample number, $n$ is the number of rows (or columns) in the spectrum data matrix, $S_s$ stands for scene spectrum, $S_B$ denotes the spectrum to be assessed (i.e., $S_O$ or $S_S''$), and $i$, $j$ are the row and column indices of the spectrum data matrix.

$$\text{B} = \frac{1}{2000} \sum_{p=1}^{2000} R_p \tag{17}$$

where $p$ is the sample number, and $R$ is the RMSE for the amplitude spectrum calculated with (14).

$$\text{C} = \frac{1}{2000} \sum_{p=1}^{2000} \frac{1}{n^2} \sum_{i=1}^{n} \sum_{j=1}^{n} \left( T_{s,p,ij} - T_{B,p,ij} \right) \tag{18}$$

where $p$ is the sample number, $n$ is the number of rows (or columns) in the spectrum data matrix, $T_s$ stands for scene BT, $T_B$ denotes the BT to be assessed (i.e., observed BT or SE BT), and $i$, $j$ are the row and column indices of the spectrum data matrix.

$$\text{D} = \frac{1}{2000} \sum_{p=1}^{2000} E_p \tag{19}$$

where $p$ is the sample number, and $E$ is the RMSE of BT calculated with (15).

**Table 4.** The average amplitude spectrum error for the entire test dataset generated with the scenes chosen from 89-GHz AMSR2 BTs.

|  | Average Error | Average RMSE |
|---|---|---|
| $S_O$ | 275.84 | 392.66 |
| $S_S''$ | 231.52 | 307.80 |

**Table 5.** The average reconstruction error for the entire test dataset generated with the scenes chosen from 89-GHz AMSR2 BTs.

|  | Average Error (K) | Average RMSE (K) |
|---|---|---|
| Observed BT | $-3.61 \times 10^{-4}$ | 5.24 |
| SE BT | $4.15 \times 10^{-5}$ | 4.05 |

The results obtained from processing the test dataset demonstrate that the SE method has the capacity to enhance the spatial resolution of the real-aperture microwave radiometer while simultaneously improving the accuracy of BT values.

## 4. Satellite Data Processing

### 4.1. Dataset Generation

In order to achieve the best possible results when processing the observed BTs from SMR's 6.925-H channel using SE-CNN, the features of the training dataset need to closely resemble the real scenes. This means that the selected scene BTs, denoted $T_S$, should closely approximate the actual distribution of BTs for the scene at 6.925 GHz in the H-polarization.

Therefore, following the steps outlined above in Section 2.2, the dataset is generated with careful consideration. During Step 1, observed BTs from the AMSR2 [21] 6.9-H channel (L1B product) are chosen as the simulated scene, denoted $T_S$. Additionally, the antenna main beam pattern of SMR's 6.925-H channel is selected as $G$, as depicted in Figure 12a. In Step 4, a matrix size of $n = 75$ is adopted, meaning that both the input and output of the SE-CNN are matrices of dimensions $75 \times 75$.

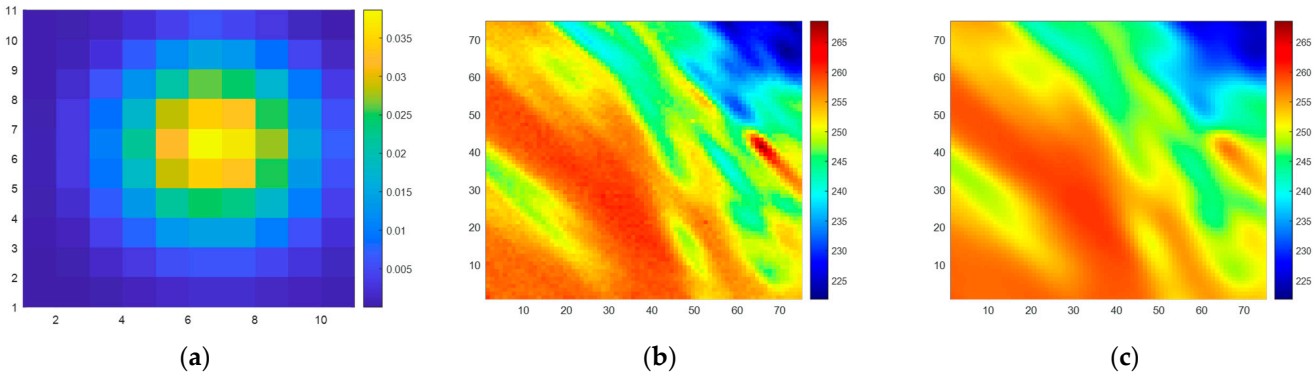

(a)                          (b)                          (c)

**Figure 12.** An example of scene observation with the scene chosen from 6.9-GHz AMSR2 BTs: (**a**) antenna main lobe pattern; (**b**) scene chosen from 6.9-GHz AMSR2 BTs; (**c**) observed BT.

A training dataset consisting of 8000 samples and a testing dataset containing 2000 samples are constructed. As depicted in Figure 12b,c, an example of the original scene BT $T_S$ and the observed BT $T_G$ is shown. The corresponding sample in the dataset, including the network input ($S_D'$) and target output ($S_D$), is illustrated in Figure 13.

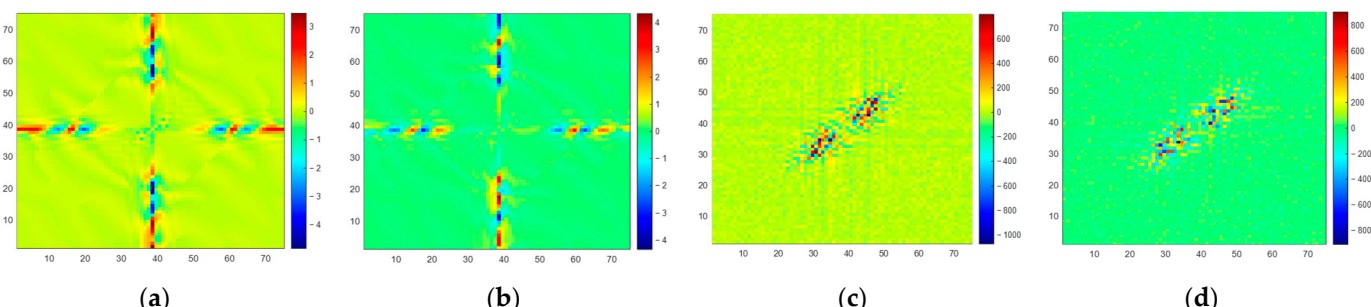

**Figure 13.** An example of network input ($S'_D$) and target output ($S_D$) with the scene chosen from 6.9-GHz AMSR2 BTs: (**a**) real part of $S'_D$; (**b**) imaginary part of $S'_D$; (**c**) real part of $S_D$; (**d**) imaginary part of $S_D$.

*4.2. Hyperparameter Determination*

In this subsection, the Bayesian search algorithm [27] is employed to search through more than 40 sets of hyperparameter combinations. To expedite the hyperparameter search, an early stopping strategy known as the Hyperband algorithm [28] is utilized to terminate the training of poorly performing neural network models. Following the search, the hyperparameters yielding the best convergence of the neural network are selected. The network structure parameters are outlined in Table 6. Additionally, the input matrix dimension is set to $n = 75$, batch size to 64, and each convolutional layer employed a $5 \times 5$ kernel. The initial learning rate is set to 0.063, and the configuration parameters for the Adam optimizer are set as $\beta_1 = 0.514$, $\beta_2 = 0.686$, and $\varepsilon = 4.540 \times 10^{-8}$. The corresponding convergence curve of the neural network is illustrated in Figure 14.

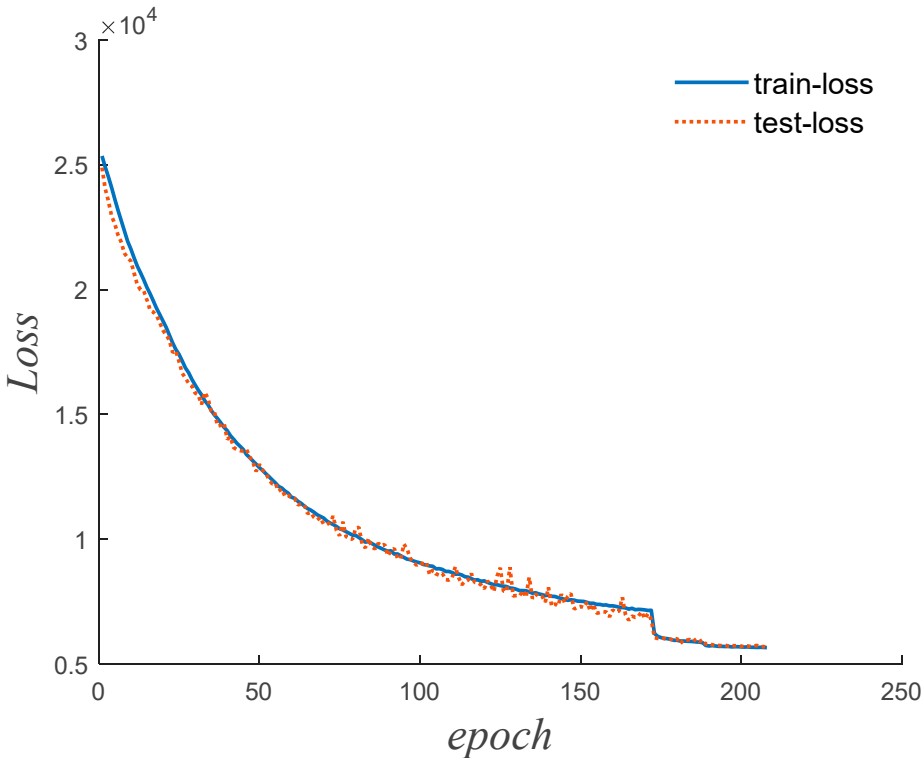

**Figure 14.** Convergence curve of SE-CNN trained by the dataset generated with the scenes chosen from 6.9-GHz AMSR2 BTs.

**Table 6.** Network structure parameters for SE-CNN trained by the dataset generated with the scenes chosen from 6.9-GHz AMSR2 BTs.

|  | Layer | Structure Parameter | Value |
|---|---|---|---|
| **Main branch** | FC1 | Node number $(n^2)$ | 5625 |
|  | FC2 | Node number $(2n^2)$ | 11,250 |
|  | Dropout | Probability (dropout) | 0.170 |
|  | C1 | Kernel number $(m_1)$ | 16 |
|  | C2 | Kernel number $(m_2)$ | 32 |
|  | C3 | Kernel number | 2 |
| **Side branch** | BC1 | Kernel number $(m_4)$ | 16 |
|  | BC2 | Kernel number $(m_5)$ | 32 |
|  | BC3 | Kernel number | 2 |

*4.3. Test Dataset Processing Result*

After the training of SE-CNN is completed, the samples from the test dataset are used to evaluate the performance of the neural network. Several test samples are randomly selected from the test dataset, and their corresponding test results are showcased. As depicted in Figure 15, three randomly chosen test samples along with their respective test outcomes are presented. In Figure 15a,d,g, the BTs of three simulated scenes (Scene 1, Scene 2, and Scene 3) are displayed, while Figure 15b,e,h showcase the corresponding observed BTs. Additionally, Figure 15c,f,i portray the reconstructed BTs achieved through the SE method (referred to as SE BTs). From Figure 15, it can be observed that the observed BTs, obtained by performing a weighted average with the antenna pattern, appear blurred and smooth, resulting in the loss of fine details. This implies a reduction in spatial resolution. However, after undergoing SE processing, the reconstructed BTs possess richer detail information and are much closer to the original simulated scene's BTs. Consequently, the spatial resolution is improved.

For a more intuitive presentation of the test results, a cross-section is randomly selected from each of the three scenes, and the corresponding one-dimensional BT curves are shown in the figures. The one-dimensional BT curve at the 18th column of the Scene 1 is displayed in Figure 16a, the corresponding curve at the 50th column for Scene 2 is represented in Figure 16b, and the curve at the 52nd column for Scene 3 is illustrated in Figure 16c. From Figure 16, it is evident that the SE BT provides a more accurate representation of the original scene's BT distribution compared to the observed BT, especially in areas with rapid changes. Overall, the SE BT curve aligns more closely with the original scene's BT curve. This implies that through the SE method, not only is the spatial resolution enhanced, but also the accuracy of BT values is improved.

From the perspective of spectrum error, the amplitude spectrum errors of observed BT and SE BT are illustrated in Figure 17 (no apparent pattern was found in phase spectrum error, so it is not discussed). In Figure 17a,c,e, it can be seen that some spatial frequency components in the observed spectrum (i.e., $S_O$) have significant errors, while in the spectrum of SE BT (i.e., $S_S''$), errors in the corresponding spatial frequency components are significantly reduced, as shown in Figure 17b,d,f. This figure demonstrates that the SE method can recover some spatial frequency components in the scene's spectrum, including high-frequency components. This is also the principle behind how the SE method can enhance spatial resolution.

Regarding quantitative analysis, the spectrum error and the BT error are calculated. The RMSE for the amplitude spectrum of observed BTs is denoted $R_O$, and the RMSE for the amplitude spectrum of SE BTs is denoted $R_S''$ (see (14)). After calculation, the $R_O$ values for Scene 1, Scene 2, and Scene 3 are 166.22, 328.04, and 567.50, respectively. The corresponding $R_S''$ values are 80.25, 168.21, and 208.95, representing reductions of 51.72%, 48.72%, and 63.18%, respectively, as shown in Table 7. The RMSE for observed BTs is denoted $E_O$, and the RMSE for SE BTs is denoted $E_S''$, (see (15)). After calculation, the $E_O$ values for Scene 1, Scene 2, and Scene 3 are 2.22 K, 4.37 K, and 7.57 K, respectively. The corresponding $E_S''$ are 1.03 K, 2.17 K, and 2.67 K, representing reductions of 53.60%, 50.34%, and 64.73%, respectively, as shown in Table 8.

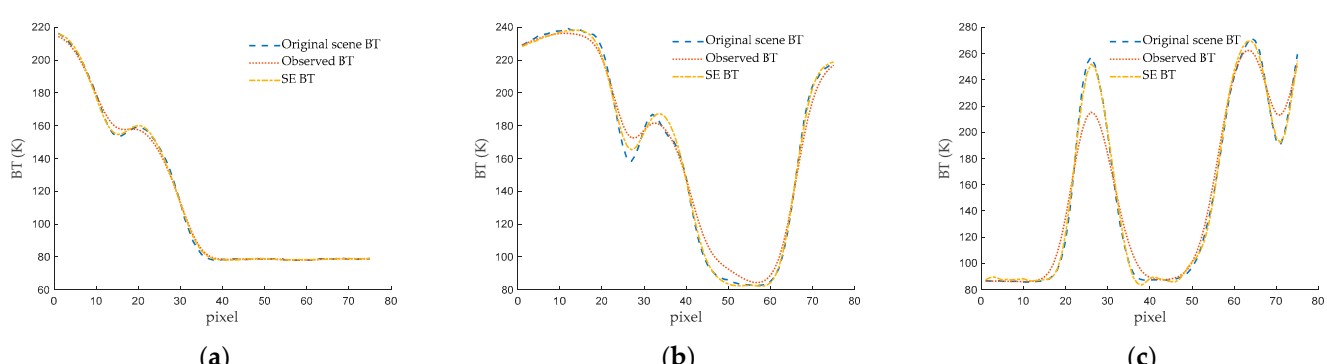

**Figure 15.** Processing results of the test dataset samples. (**a**,**d**,**g**) BT images of simulated scenes, Scene 1, Scene 2, and Scene 3; (**b**,**e**,**h**) corresponding observed BT images; (**c**,**f**,**i**) corresponding SE BT images.

**Figure 16.** One-dimensional BT curves of the three scenes shown in Figure 15: (**a**) Scene 1; (**b**) Scene 2; (**c**) Scene 3.

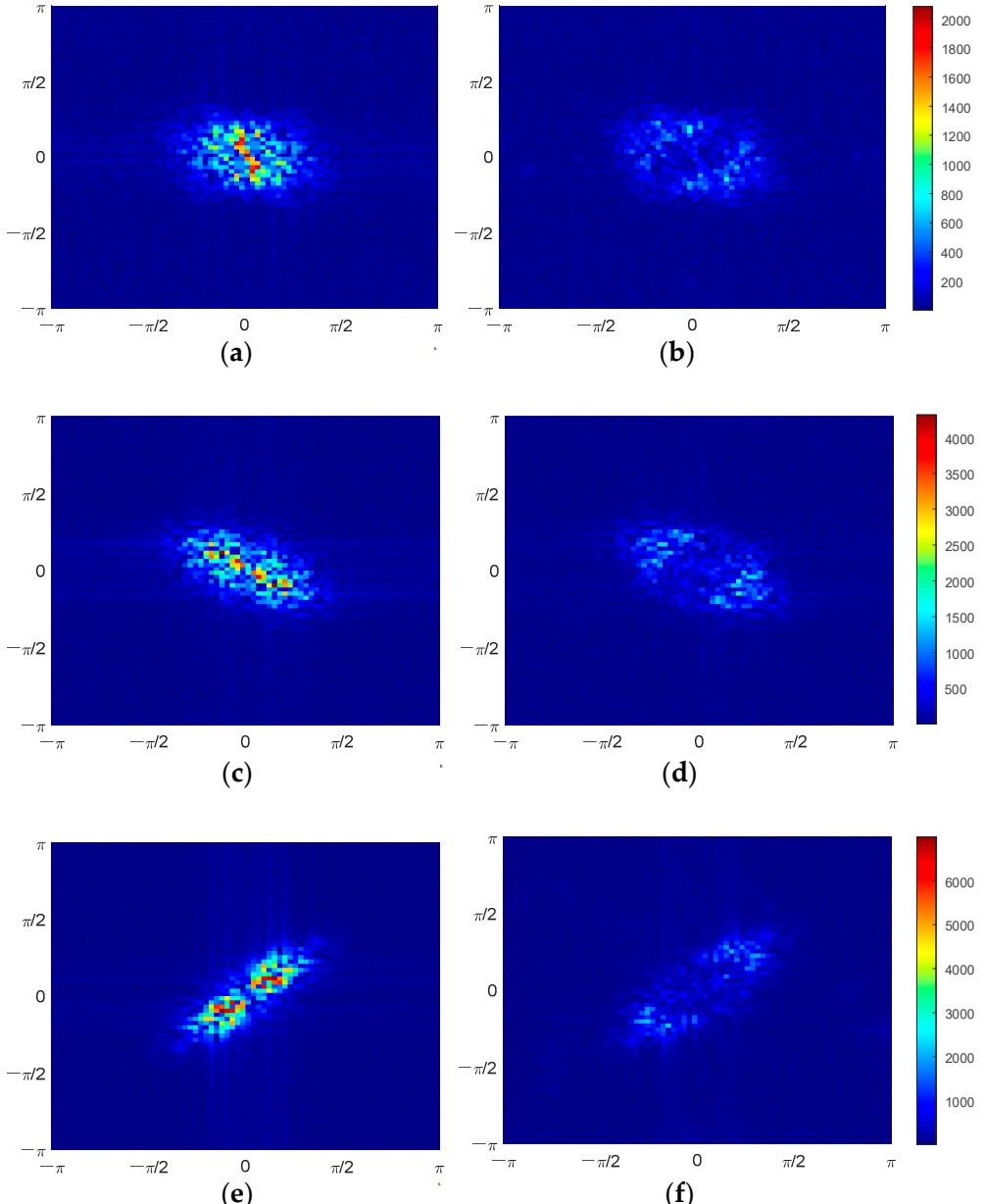

**Figure 17.** Amplitude spectrum error. (**a**,**c**,**e**) RMSE in $S_O$ of Scene 1, Scene 2, and Scene 3; (**b**,**d**,**f**) RMSE in $S_S''$ of Scene 1, Scene 2, and Scene 3.

**Table 7.** Amplitude spectrum error of observed BT and SE BT for Scene 1, Scene 2, and Scene 3.

|  | $R_O$ | $R_S''$ | Reduction |
|---|---|---|---|
| Scene 1 | 166.22 | 80.25 | 51.72% |
| Scene 2 | 328.04 | 168.21 | 48.72% |
| Scene 3 | 567.50 | 208.95 | 63.18% |

**Table 8.** Error of observed BT and SE BT for Scene 1, Scene 2, and Scene 3.

|  | $E_O$(K) | $E_S''$(K) | Reduction |
|---|---|---|---|
| Scene 1 | 2.22 | 1.03 | 53.60% |
| Scene 2 | 4.37 | 2.17 | 50.34% |
| Scene 3 | 7.57 | 2.67 | 64.73% |

To further validate the effectiveness of the SE method, the tests on all 2000 samples from the test dataset is conducted. The average amplitude spectrum errors of $S_O$ and $S_S''$ for all test samples are computed, as shown in Table 9. The average error (refer to (16)) of $S_O$ is 59.72, while the average error of $S_S''$ is 41.45, showing a decreasing of approximately 30.59%; the average RMSE (refer to (17)) of $S_O$ is 157.23, whereas the average RMSE of $S_S''$ is 79.58, expressing a reduction of approximately 49.39%. The average error and the average RMSE of observed BT and SE BT for all test samples are calculated, respectively, as shown in Table 10. The average error (refer to (18)) for both observed and SE BT is around 0 K. The average RMSE (refer to (19)) for observed BT is 2.10 K, while the average RMSE for SE BT is 1.03 K, representing a reduction of approximately 50.95%. The improved spatial resolution of SE BT brings it closer to the original BT, resulting in smaller reconstruction error.

**Table 9.** The average amplitude spectrum error for the entire test dataset generated with the scenes chosen from 6.9-GHz AMSR2 BTs.

|  | **Average Error** | **Average RMSE** |
|---|---|---|
| $S_O$ | 59.72 | 157.23 |
| $S_S''$ | 41.45 | 79.58 |

**Table 10.** The average reconstruction error for the entire test dataset generated with the scenes chosen from 6.9-GHz AMSR2 BTs.

|  | **Average Error (K)** | **Average RMSE (K)** |
|---|---|---|
| Observed BT | $1.2 \times 10^{-3}$ | 2.10 |
| SE BT | $-1.68 \times 10^{-4}$ | 1.03 |

The processing results of the test dataset indicate that the SE method can enhance the spatial resolution of the real-aperture microwave radiometer while also improving the accuracy of BT values.

### 4.4. Satellite Data Processing Result

The ultimate goal of the SE method is to enhance the spatial resolution of satellite-borne radiometer. In this section, the observed BT from the 6.925-H channel of SMR is used to test the performance of the SE method. Some samples of observed BT data are randomly selected from SMR's observations during November–December 2018, January 2019, and May 2019. These selected data samples are then processed using the trained SE-CNN model, and the corresponding SE BTs are shown in Figure 18.

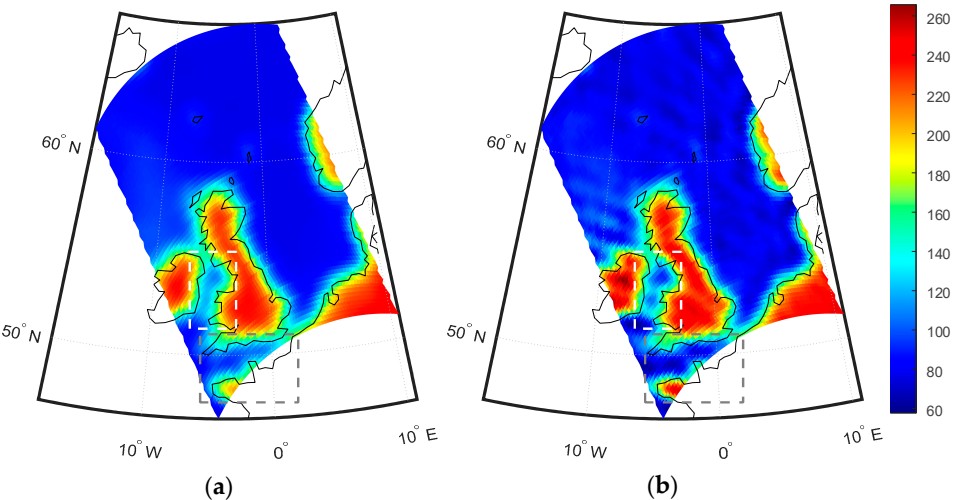

**Figure 18.** *Cont.*

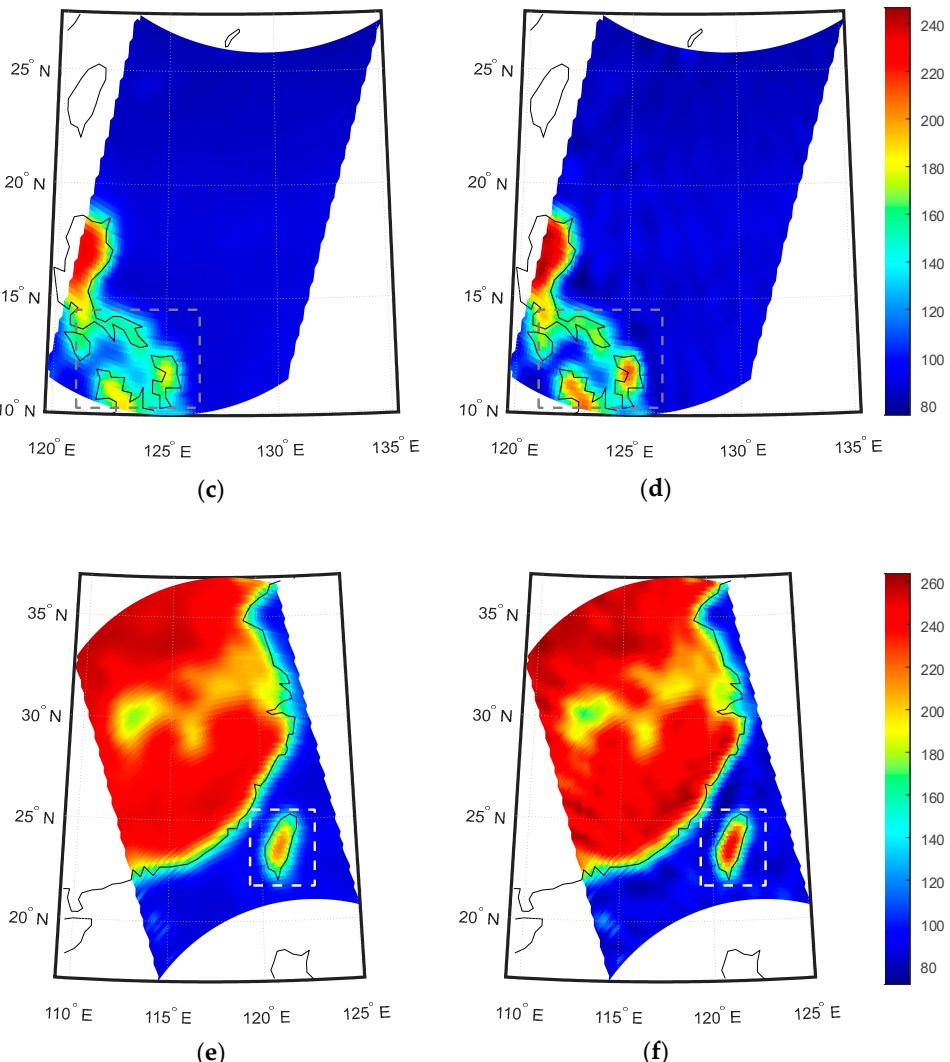

**Figure 18.** Processing results of satellite-observed BTs. Panels (**a**,**c**,**e**) show the observed BT from SMR, while panels (**b**,**d**,**f**) display the corresponding SE BT.

From Figure 18, it can be observed that the BT's spatial resolution is enhanced and land-to-sea contamination is mitigated through the SE method. The observed BTs from SMR exhibit significant contamination due to the lower observation resolution. In coastal areas, the observed BTs are higher than in open ocean regions. Moreover, the BTs in sea islands and peninsulas are lower than those in inland areas, as shown by the dashed boxes in Figure 18a,c,e. However, after undergoing the SE method, as depicted in Figure 18b,d,f, there is a noticeable improvement in the land-to-sea contamination. Specifically, the BTs in coastal regions are more consistent with those in open ocean regions, and the BTs in sea islands and peninsulas are closer to those in inland areas. This indicates an enhancement in the accuracy of BT values.

Furthermore, after the application of the SE method, the transition zones in the coastal areas in Figure 18b,d,f become narrower. To provide a clearer depiction of these changes, one-dimensional BT profiles are extracted randomly from SMR observed BT and SE-enhanced BT data matrices of the three scenes. These profiles are illustrated in Figure 19. It can be observed that the transition zones in the land-to-sea boundary area become narrower. Additionally, the narrow land areas exhibit BTs that are more consistent with those in the inland regions, while the narrow ocean areas display BTs more akin to the open ocean regions. These results collectively demonstrate the enhancement in spatial resolution and accuracy of BT values achieved through the SE method.

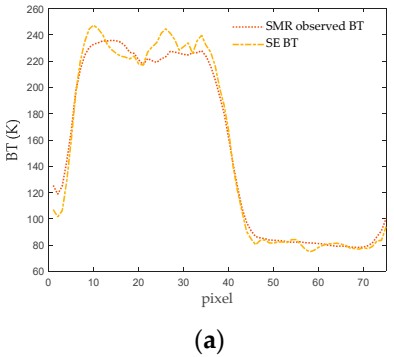
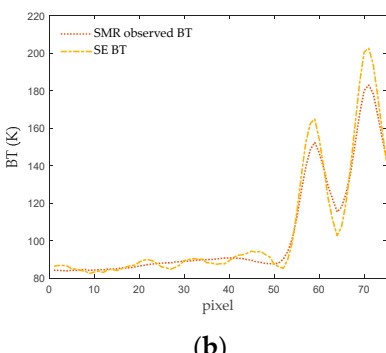
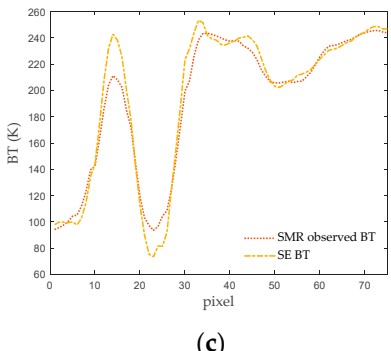

(**a**)    (**b**)    (**c**)

**Figure 19.** One-dimensional cross-sectional BT profiles are shown in (**a**–**c**), corresponding to the three scenes in Figure 18.

## 5. Discussion

The SE method is utilized to enhance the spatial resolution of the microwave radiometer and also improve the accuracy of BT values. When employing the SE method, there are several points to be aware of.

When selecting scene BT to generate the dataset, it is crucial to ensure that the frequency of the chosen scene BT matches the frequency of the satellite-observed BT that needs to be processed. For instance, if the frequency of the satellite-observed BT is at 6.9 GHz, then the frequency of the original scene BT to generate dataset should also be at 6.9 GHz. When frequencies differ, the data used to train the neural network and the data to be processed will have different characteristics, which can result in larger errors in the final processing results.

Furthermore, the scene BTs used to generate the dataset should closely resemble the actual distribution of scene BTs. This requires a sufficient number of scenes in the dataset to cover a wide range of scene types. What is equally important is that the clarity (spatial resolution) of the scene BT images used to generate the dataset is sufficiently high. This ensures that they accurately reflect the distribution characteristics of real scene BTs.

Considering these factors, AMSR2-observed BTs are chosen as the source of original scene BTs for generating the dataset. However, the spatial resolution of AMSR2's observed BT images is not very high. If higher-resolution BT images were available, the performance of the SE method could potentially be further enhanced.

## 6. Conclusions

The antenna of a real-aperture microwave radiometer acts as a low-pass filter, suppressing the high-frequency components of scene BT's spectrum. This results in a reduction of the observed BT's spatial resolution. In this paper, the observed BT's spectrum is extended by the proposed SE method, making it more closely resemble the true scene BT's spectrum. As a result, the processed BT becomes more aligned with the actual scene BT.

Unlike traditional methods that primarily focus on improving spatial resolution while paying less attention to the accuracy of BT values, the SE method not only enhances spatial resolution but also improves the precision of BT values. This means that, in contrast to conventional approaches that may prioritize achieving finer details, the SE method simultaneously enhances spatial resolution and ensures that the reconstructed temperature values are more accurate, resulting in a more comprehensive and precise representation of the observed data.

The SE method utilizes frequency domain transformation and computations to restore the spectrum of the original scene's BT. It then employs a convolutional neural network to correct errors present in the calculated scene spectrum. Subsequently, it reconstructs a higher-spatial-resolution BT image based on this spectrum, thereby enhancing the spatial resolution of the real-aperture microwave radiometer.

The neural network needs to undergo training on the dataset to become effective. When generating the training dataset, the ideal values and calculated values of the difference between the scene spectrum and the observed spectrum are used as the training target and training input, respectively. This processing enables the neural network to converge more effectively and quickly during training. The generated dataset's scene BT should closely resemble the real scene, and its frequency should match that of the satellite-observed BT to be processed.

The performance of the SE method is validated through simulation data and satellite-measured data. In the simulation tests, three random scenes are selected. The results indicate that compared to observed BT, SE BT exhibits clearer details in variations, with a visibly enhanced spatial resolution. With the increased spatial resolution, the SE BT errors for these three scenes are reduced compared to the observed BT errors. Further calculations are performed on the entire test dataset, showing that the average RMSE of SE BTs is reduced compared to observed BTs.

For satellite observation data testing, data from the 6.925-H channel of the SMR instrument aboard the HY-2B satellite is used. Three scenes are randomly selected for processing with the SE method. The results demonstrated that the transition zones between land and ocean exhibit narrower BT gradients after SE processing, indicating an enhancement of spatial resolution. Moreover, effective improvement in mitigating land-to-sea contamination is observed, with islands and peninsulas showing BTs closer to those of the inner land, and narrow ocean areas showing BTs closer to those in open ocean regions. This outcome indicates that the enhancement of spatial resolution also leads to an increase in the accuracy of BT data.

**Author Contributions:** Conceptualization, G.Z. and C.X.; methodology, simulation and validation, G.Z.; data curation, Y.H., Z.C. and W.W.; writing—original draft preparation, G.Z.; writing—review and editing, G.Z. and C.X. All authors have read and agreed to the published version of the manuscript.

**Funding:** This research received no external funding.

**Data Availability Statement:** The data used to support the research are available from the websites https://gportal.jaxa.jp (AMSR2 data, accessed in 30 April 2021) and https://osdds.nsoas.org.cn (SMR data, accessed in 30 April 2021).

**Acknowledgments:** The authors would like to thank the National Satellite Ocean Application Service (NSOAS) for providing data, and would also like to thank the Japan Aerospace Exploration Agency (JAXA) for providing data.

**Conflicts of Interest:** The authors declare no conflict of interest.

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
