# Peer review of "Spectrum Extension of a Real-Aperture Microwave Radiometer Using a Spectrum Extension Convolutional Neural Network for Spatial Resolution Enhancement"

_remotesensing, doi:10.3390/rs15245775_

Round 1
Reviewer 1 Report
Comments and Suggestions for Authors
In general this work presents a semi-novel approach to enhancing microwave radiometer observations from conical remote sensing scanning systems. It mixes traditional FFT based optimal filtering and the use of a Talyor series expansion with an CNN solution to minimize loss of high frequency spatial information. While I believe the method may have merit, it is somewhat hard to tell from the presentation, that provide minimal quantitative support for its claims. Through out this work, among other thing very idealized main lobe/PSF is used, which limits the understanding of it application to real-world microwave imagery data.
My major concerns with this work are two fold. First I think in the simulated world one should start with much higher resolution imagery, to explore the real impact or gain from this technique to the overall processing of MW data. As it stands we are sort of saying we can't do as well as existing instrument but may be better then we are now, not how does it represent the truth. Second is that the overall change in BT being used as a quantitative metric seems to miss the mark of have we enhanced the high frequency spatial information. The work may need to answer the question of, in the Fourier space what it the error in various spatial frequency domains both from an amplitude and phase perspective.
Finally, the processed satellite data seem to provide little quantitative information. Do the results better match AMSR. How can one tell if the results better represent the real scene. What are the artifact introduces, and do the artifact effects introduces out weigh the gains? Some simple difference images might help illustrated this. These questions should be address in any resubmission. The authors also need to state what type of AMSR data they are comparing against. Has BG been applied or is it just raw data.
That being said, this is a reasonably written work, that could however benefit from a good proofing.
I have place specific notes in the attached version of the paper.

Comments on the Quality of English Languagesee comments above.
Reviewer 2 Report
Comments and Suggestions for Authors
The authors present the Spectrum Enhancement (SE) method which could extend the observed BT spectrum. Moreover, the authors use a neural network to correct errors in the calculated spectrum. This method effectively enhances the spatial resolution of real aperture microwave radiometers and concurrently improves the accuracy of BT values. In general, I think this manuscript can be considered as potential publication in Remote Sensing after addressing some issues.
1. At the end of the introduction, the article proposes that the radiometer only has a vertical polarization (V-pol) channel at 23.85GHz, while other frequencies have both vertical polarization and horizontal polarization (H-pol) channels. In the end, the author chose 6.925-H. Please add the reason for choosing this frequency and polarization.
2. The format of this paper should be carefully double-checked. Some minor mistakes can be found in the manuscript, for example, the vertical coordinates and drawing style of Figure 10 and the horizontal coordinates of Figure 12 are not unified, etc.
3. The average error is mentioned in Table 3 of the article, but the definition of average error is not provided in the entire text. After using the SE method to process, the average error of BT shows a negative value. Please explain the meaning of the negative value here and the reasons for its occurrence.
4. The SE method proposed in the article improves the accuracy of BT values, but how does it compare to other existing methods? Please provide additional explanation.
Comments on the Quality of English LanguageThe English expression is relatively complete.
